

# Full vaccination coverage and associated factors among children aged 12 to 23 months in remote rural area of Demba Gofa District, Southern Ethiopia

Tadele Dana Darebo[1], Bahru Belachew Oshe[2] and Chala Wegi Diro[1]

[1] School of Public Health, Wolaita Sodo University, Wolaita Sodo, South Ethiopia
[2] Maternal and Child Health Department, Gofa Zone Health Office, Gofa, South Ethiopia

## ABSTRACT

**Background:** Full vaccination refers to the administration of vaccines/antigens recommended for children in the first year of life. However, little is known about full vaccination in remote, rural Ethiopia. This study aimed to measure full vaccination coverage and associated factors among children aged 12 to 23 months in Demba Gofa District, Southern Ethiopia.

**Methods:** A community-based cross-sectional study was conducted in April and May 2019 using a multistage sampling technique to select 677 mothers with children 12–23 months of age. Data was collected using a pre-tested structured questionnaire, and data were edited, coded, entered, and cleaned using Epi Info v3.1 and analyzed using SPSS v20. Bivariate and multivariable logistic regression was used to understand associations between dependent and independent variables.

**Results:** Three-hundred and nine children (47.0%) were fully vaccinated, 274 (41.7%) were partially vaccinated, and 74 (11.3%) were not vaccinated at all. Children were more likely to be vaccinated if decisions were made jointly with husbands (AOR = 1.88, 95% CI [1.06–3.34]), were made by mothers (AOR = 4.03, 95% CI [1.66–9.78]), followed postnatal care (AOR = 5.02, 95% CI [2.28–11.05]), if the child's age for completing vaccination was known (AOR = 2.54, 95% CI [1.04–6.23]), and if vaccinations did not make the child sick (AOR = 0.32, 95% CI [0.16–0.64]).

**Conclusion:** Full vaccination coverage was less than average in the study district and far below the governmental target (90%) necessary for sustained control of vaccine-preventable diseases. Interventions targeted towards maternal healthcare decision-making, postnatal care, knowledge on vaccination timing, and importance should be prioritized to improve full vaccination coverage. A continuous supply of vaccination cards needs to be ensured to improve vaccination conditions.

## INTRODUCTION

Vaccination is the efficient way to protect people from infectious diseases by stimulating the host immune system to produce antibodies or other specific immune defenses to protect against particular infectious diseases (*Neiburg & Nancy, 2011*; *UNICEF, 2013*).

Corresponding author
Tadele Dana Darebo,
danatadele@gmail.com

Vaccination has reduced and eliminated various childhood diseases globally, including diphtheria, tetanus, pertussis, polio, measles, and tuberculosis. Several new vaccines have now been added to public health programs (hepatitis B, *Haemophilus influenzae* B (HiB), pneumococcus, and rotavirus) (*Herliana & Douiri, 2018*; *WHO, 2015*; *Negussie et al., 2016*; *Federal Ministry of Health of Ethiopia, 2012*; *Gentile, 2010*; *Mbengue et al., 2017*; *Gualu & Dilie, 2017*; *Federal Ministry of Health, Addis Ababa, 2015*). Global vaccination coverage has increased, contributing to decreases in child mortality from 9.6 million in 2000 to 5.9 million in 2015, especially in low and middle-income countries. Global vaccination coverage has increased, which has contributed to decreases in child mortality from 9.6 million in 2000 to 5.9 million in 2015 across all regions of the world, especially low and middle-income countries (LMICs) (*Herliana & Douiri, 2018*; *Tefera et al., 2018*; *Sk et al., 2018*; *Holipah, Maharani & Kuroda, 2018*; *Yenit, Gelaw & Shiferaw, 2018*; *Aregawi et al., 2017*; *Meleko, Geremew & Birhanu, 2017*; *Acharya et al., 2018*; *Ekouevi et al., 2018*).

The World Health Organization (WHO) recommends, which is also strictly endorsed by Ethiopia, that a child is fully vaccinated if he/she has the BCG (tuberculosis) vaccine at birth; oral polio vaccine (OPV) at birth, 6, 10, and 14 weeks; pentavalent vaccine for diphtheria, tetanus, pertussis (whooping cough), hepatitis B, and HiB at 6, 10, and 14 weeks; pneumococcal conjugate vaccine (PCV) at 6, 10, and 14 weeks; rotavirus at 6 and 10 weeks; inactivated polio vaccine (IPV) at 14 weeks; and measles at 9 months (*WHO, 2015*; *Negussie et al., 2016*; *Federal Ministry of Health of Ethiopia, 2012*). The Expanded Program on Immunization (EPI) agenda plans by applying WHO standards to give the primary vaccination series to at least 90% of children. Despite these efforts, the target has still to be attained in many LMICs, but that target has still to be attained in many LMICs (*Negussie et al., 2016*; *Bekele et al., 2017*; *MMWR Morb Mortal Wkly Rep, 2019*; *Central Statistical Agency: Addis Ababa Ethiopia, 2017*; *Central Statistical Agency: Addis Ababa Ethiopia, 2019*; *Adeleye & Mokogwu, 2016*; *Awoh & Plugge, 2015*; *Feldstein et al., 2017*). Despite this effort over 20,000 children die each day from preventable infectious diseases, and a larger part of these deaths occur in sub-Saharan Africa (SSA) and South East Asia (SEA) compared to the rest of the world (*Gentile, 2010*; *Gualu & Dilie, 2017*; *Sheikh et al., 2018*; *Farzad et al., 2017*; *Ganguly et al., 2018*; *Mihigo et al., 2017*).

Globally, 22.6 million infant children are partially protected through vaccination and over 25% of these are reported to be in LMICs (*Yenit, Gelaw & Shiferaw, 2018*; *Aregawi et al., 2017*; *Meleko, Geremew & Birhanu, 2017*). An estimated 2.5 million children aged less than 5 years die annually due to diseases that can be prevented by vaccination (*Tefera et al., 2018*; *Aregawi et al., 2017*; *Adokiya, Baguune & Ndago, 2017*). About 14.8 million children who are not vaccinated with pentavalent 3 before celebrating their first year are found only in only 10 countries, including Ethiopia (*Yenit, Gelaw & Shiferaw, 2018*; *Aregawi et al., 2017*; *Mihigo et al., 2017*; *Adokiya, Baguune & Ndago, 2017*; *Mohamed et al., 2016*). Despite the fact that most under-five deaths can be easily tackled by vaccination, about half occur in SSA (*WHO, 2015*; *Federal Ministry of Health of Ethiopia, 2012*; *Adedire et al., 2017*; *Shemwella et al., 2017*; *Kassahun, Biks & Teferra, 2015*; *Debie & Taye, 2014*; *Mohamud et al., 2014*; *Legesse & Dechasa, 2015*; *Ebrahim & Salgedo, 2015*;

*Facha, 2015*). The most common vaccine preventable diseases that result in morbidity and mortality under-five children are pneumonia, diarrheal diseases and measles. More than three million cases of pneumonia is reported each year claiming the lives of about 20% of an estimated 40,000 annual deaths. Diarrehal diseases follows accounting 1.7 million cases per year and measles being the third with the 50 per one million population per year with the case fatality rate of 3–6% (*Nour et al., 2020*; *Konwea, David & Ogunsile, 2018*; *Kiptoo et al., 2015*).

Ethiopia aimed to target at least 90% of the population with all vaccines by 2020. However, studies conducted in Ethiopia in children aged 12–23 months have reported vaccination rates from 39% to 43%. Full vaccination coverage is highly discripant at the regional level with the lowest reported to be as low as 15% in Afar with 15% and as high as highest Addis Ababa 89%. In Addis Ababa, different factors including sociodemographic and economic factors, maternal healthcare service utilization, knowledge and attitudes of mothers/caregivers towards vaccination, accessibility, and perceived availability were identified to affect full vaccination coverage (*Central Statistical Agency: Addis Ababa Ethiopia, 2017*; *Central Statistical Agency: Addis Ababa Ethiopia, 2019*; *Facha, 2015*; *Nour et al., 2020*).

However, there is still a lack of information on the prevalence and associated factors of full vaccination coverage across different regions of Ethiopia and areas of the country where epidemics intermittently occur. This study therefore aimed to measure full vaccination coverage and associated factors among children aged 12 to 23 months in Demba Gofa District, Southern Ethiopia to assist program implementers and frontline health workers understand the factors influencing full vaccination coverage and to plan for default tracing mechanisms.

## MATERIALS AND METHODS

### Study setting and design

A community-based cross-sectional study was conducted in April and May 2019 in the Demba Gofa District, Ethiopia, which is located in Southern Ethiopia 514 km from Addis Ababa. The district contains 34 kebeles, all rural, with a total population of 96,427; 1,659 (1.72%) were children aged 12–23 months. Demba Gofa District has four governmental health centers, 34 health posts, and 15 private primary clinics. The EPI service was provided only by governmental health facilities (*Demba Gofa District Health Office, 2018*).

### Patient and public involvement

Not applicable for this study.

### Source population

The source population was all children aged 12–23 months paired with their mothers/caregivers.

### Inclusion/exclusion criteria

The inclusion criteria were all mothers/caretakers caregiverscaregiverswith children aged 12–23 months who were residents of selected kebeles (for at least 6 months) in the district, while exclusion criteria were mentally disabled/critically ill mothers/caretakers.

### Sample size determination

The sample size was determined using the single population proportion formula and P = 27.7% as the prevalence of full immunization coverage among children aged 12–23 months in Ambo Woreda, Central Ethiopia (*Etana & Deressa, 2012*). By using a 95% level of significance, a margin of error of 5%, a design effect of 2, and a non-response rate of 10%, the final sample size was 677:

$$n = (Z\alpha/2)2p(1-P)$$

$$d2$$

### Sampling procedure

Multi-stage sampling was used to select the study population at the community level. In the first stage, 10 kebeles were selected using a simple random sampling method from a total of 34 kebeles in the district. The total sample size was divided by the size of all households that had children aged 12–23 months in each selected kebele. Then, children the total sample size were was proportionally allocated according to the number of children they had in each kebele. Finally, a simple random sampling method was employed to select children using the lottery method from each kebele. When there were two or more children in the same household, the lottery method was used to select only one. When there was no eligible child in the selected household, the next household was included in the study. If eligible participants were not at home during data collection, interviewers revisited the households twice.

### Data collection and quality control

Data were collected using a pre-tested and structured questionnaire developed from the related literature used to conduct similar study in elsewhere (*Mbengue et al., 2017*; *Gualu & Dilie, 2017*; *Federal Ministry of Health, Addis Ababa, 2015*; *Tefera et al., 2018*). The questionnaire was initially prepared in English and then translated to Gofatho (local Gofa language) and then translated back into English by the qualified translators to check for consistency. Data were collected through face-to-face interviews from mothers/caregivers. Training was given for 2 days before data collection by the principal investigator to the data collectors and supervisors on the purpose of the study, data collection tools, and on minimizing the recall bias. Mothers/caregivers were asked to show vaccination cards (if available, the interviewer copied the vaccination dates), and for those who had no or lost vaccination cards, mothers/caregivers were asked about the vaccination status of their children using different recall mechanisms such as asking the route of administration, injection sites, examining the BCG scar, etc. To assure data quality, data collectors and supervisors were given 2 days of training. Pretesting was

performed in 5% (*Mbengue et al., 2017*; *Adeleye & Mokogwu, 2016*) of the sample size in kebeles outside of the study area after which necessary modifications were undertaken. Daily check-ups were performed for data completeness, accuracy, and consistency.

## Operational definitions

Fully vaccinated: A child aged 12–23 months who received one dose of BCG, at least three doses of OPV, three doses of pentavalent, three doses of PCV, two doses of Rota, and one dose of measles vaccines before celebrating the first birth year (*Federal Ministry of Health, Addis Ababa, 2015*).

Partially vaccinated: If a child missed at least one dose of the 10 vaccines mentioned above (*Federal Ministry of Health, Addis Ababa, 2015*).

Unvaccinated: A child who had not received any of the 10 vaccines (*Federal Ministry of Health, Addis Ababa, 2015*).

Knowledge of vaccination: A mother/caregiver answering a mean or above of knowledge questions was considered to have good knowledge, while a mother/caregiver answering below a mean of knowledge questions was considered to have poor knowledge (*Mbengue et al., 2017*; *Yenit, Gelaw & Shiferaw, 2018*).

Attitude towards vaccination: When the respondent answered the mean and above of the ten attitude questions were deemed to have a positive attitude and those answering below the mean were deemed to have a negative attitude (*Mbengue et al., 2017*; *Yenit, Gelaw & Shiferaw, 2018*).

## Data management and analysis

After data collection, data were edited and coded for completeness and consistency and then entered into Epi Data v3.1 and exported to SPSS v20 for analysis. After cleaning data for inconsistencies, errors, and missing values, descriptive statistics (mean, median, SD, percent, frequency) were calculated to visualize the overall distribution of the study subjects for the variables under study. Bivariate analysis was performed to determine associations between independent and dependent variablesfull vaccination status. Multicollinearity was checked using a cutoff <10 based on the variance inflation factor (VIF) or tolerance test >0.1. The necessary assumptions of logistic regression were checked using Hosmer and Lemeshow tests to assess the fitness of the model at a $p$-value < 0.05. All explanatory variables associated with the the outcome variablefull vaccination status at $p$-value < 0.25 were selected for multivariable analysis. Finally, multivariate analysis was used to measure the degree of association between independent and outcome variables. Adjusted odds ratio (AOR) with 95% CI was used to determine statistically significant.

## Ethical considerations

Ethical approval was obtained from the Research Ethical Review Committee of the College of Health Sciences and Medicine of Wolaita Sodo University under the number CHSM/ERC/42. An official letter of cooperation was taken to the Gofa Zone Health Department and Demba Gofa District Health Office and a permission letter was then

received from each. Official letters also were written to the kebele administration and health posts. Informed written consent was obtained from particpants after explaining the aims of the study, and the confidentiality and anonymity of information was preserved.

# RESULTS

## Socio-demographic and economic characteristics of study participants

From a total of 677 mothers with children aged 12–23 months old, 657 were interviewed, a response rate of 97%. Most respondents (632, 96.2%) were mothers of the children, 77.8% were Protestant, 634 (96.5%) were married, and all were of Gofa ethnicity. The majority of mothers/caregivers (260, 39.6%) were illiterate, and most husbands (497, 75.6%) were farmers. A total of 131 (19.9%) families were grouped into the richest wealth quantile. The average family size was 5.79 with a range from 2 to 12. Over half (376, 57.2%) of the children were male, and most belonged to families with multiple children (Table 1).

## Maternal healthcare service utilization

Four hundred and sixty-eight (72.0%) mothers sought healthcare services through joint decisions with their husbands. Most respondents (545, 83.8%) had received antenatal care (ANC) during their last pregnancy, most (517, 79.5%) had attended ANC at least two or more times, and 485 (74.6%) had taken tetanus injections. Four hundred and twenty-five (65.4%) mothers gave birth to their last baby in healthcare institutions, but only 62 (9.5%) mothers attended postnatal care at least once (Table 2).

## Knowledge and attitudes of mothers/caregivers towards vaccination and vaccine-preventable disease

Of 657 total respondents, 603 (92.8%) had ever heard about vaccination and vaccine-preventable diseases. The majority of respondents (549, 91.0%) heard about vaccination and vaccine-preventable diseases from health care workers. Nearly half of the respondents (308, 47.4%) mentioned measles as a specific vaccine-preventable disease, and 210 (32.3%) mentioned tetanus. Respondents were also asked the objective of vaccinating children, the age at which a child should begin vaccination, the sessions needed to complete vaccination, and the correct age at which a child completes its vaccination program. Regarding knowledge, almost three-quarters (487, 74.0%) of study participants were considered to have a good knowledge, whereas 469 (71.4%) of study subjects were considered to have a positive attitude towards vaccination.

## Accessibility and vaccination coverage

All respondents reported that they had access to a health facility that provided vaccination services near them. Over 90% of health facilities that provided vaccination services in the Demba Gofa District were reported to be health posts. With respect to vaccination delivery strategy, 351 (53.4%) were static, 158 (24.0%) were outreach, and 148 (22.5%) were home to home.

**Table 1 Socio-demographic & economic characteristics of mothers/caregivers in Demba Gofa District, Southern Ethiopia, 2019 (N = 657).**

| Variables | Category | Frequency | Percent |
|---|---|---|---|
| Respondent | Mother | 632 | 96.2 |
| | Caregiver | 25 | 3.8 |
| Age of respondents (years) | 15–24 | 178 | 27.1 |
| | 25–34 | 331 | 50.4 |
| | >35 | 148 | 22.5 |
| Religion | Protestant | 511 | 77.8 |
| | Orthodox | 138 | 21.0 |
| | Muslim | 8 | 1.2 |
| Marital status | Married | 634 | 96.5 |
| | Others* | 23 | 3.5 |
| Level of education of respondents | Illiterate | 260 | 39.6 |
| | Can read & write | 43 | 6.5 |
| | Primary school (1–8) | 173 | 26.3 |
| | Secondary & pre. (9–12) | 120 | 18.3 |
| | Diploma & above | 61 | 9.3 |
| Occupation of respondents | Housewife | 548 | 83.4 |
| | Merchant | 67 | 10.2 |
| | Others** | 42 | 6.4 |
| Husbands' level of education | Illiterate | 211 | 32.1 |
| | Can read & write | 51 | 7.8 |
| | Primary school (1–8) | 159 | 24.2 |
| | Secondary & pre. (9–12) | 145 | 22.1 |
| | Diploma & above | 81 | 12.3 |
| Occupation of husbands | Farmer | 497 | 75.6 |
| | Merchant | 68 | 10.4 |
| | Gov't employee | 40 | 6.1 |
| | Others*** | 42 | 6.4 |
| Wealth index | Poorest | 132 | 20.1 |
| | Poorer | 131 | 19.9 |
| | Middle | 132 | 20.1 |
| | Richer | 131 | 19.9 |
| | Richest | 131 | 19.9 |
| Number of people living in the house | ≤3 | 109 | 16.6 |
| | 4–5 | 238 | 36.2 |
| | ≥6 | 310 | 47.2 |
| Sex of the child | Male | 376 | 57.2 |
| | Female | 281 | 42.8 |
| Birth order of the last child | One | 120 | 18.3 |
| | Two-three | 118 | 18.0 |
| | Four-five | 195 | 29.7 |
| | Above five | 224 | 34.1 |

(Continued)

| Variables | Category | Frequency | Percent |
|---|---|---|---|
| Age of the child (in months) | 12–14 | 222 | 33.8 |
| | 15–17 | 156 | 23.7 |
| | 18–20 | 147 | 22.3 |
| | 21–23 | 132 | 20.1 |

**Notes:**
Others* include divorced, widowed, separated, and single.
Others** include students, government workers, daily labor.
Others*** include student, daily labor.

**Table 2 Maternal healthcare service utilization in Demba Gofa District, Southern Ethiopia, 2019 (N = 657).**

| Variables | Category | Frequency | Percent |
|---|---|---|---|
| Healthcare seeking decision making | By herself | 84 | 12.8 |
| | Jointly with their husband | 473 | 72.0 |
| | Husband alone | 100 | 15.2 |
| ANC attendance | Yes | 551 | 83.9 |
| | No | 106 | 16.1 |
| Number of ANC attendance | Once | 28 | 5.1 |
| | Twice | 118 | 21.4 |
| | Three times | 193 | 35.0 |
| | Four times | 212 | 38.5 |
| Tetunus Toxiod (TT) vaccination | Yes | 490 | 88.9 |
| | No | 61 | 11.1 |
| TT vaccination status | TT 1 | 79 | 16.1 |
| | TT 2 | 257 | 52.4 |
| | TT 3 | 113 | 23.1 |
| | TT 4 | 41 | 8.4 |
| Delivery place of the last baby | Home | 227 | 34.6 |
| | Health institution | 430 | 65.4 |
| PNC attendance | Yes | 63 | 9.6 |
| | No | 594 | 90.4 |

From a total of 657 children aged 12–23 months old, under half (309, 47.0%) were fully vaccinated or completed all the recommended vaccines before celebrating their first birthday. No child had a vaccination card, and all data were collected by recall alone. The reasons mentioned for partial and unvaccinated children were absenteeism of vaccinators, time inconvenience, not knowing the exact vaccination date, beliefs that vaccination has no use, fear of side effects, and religious and cultural issues.

## Factors associated with full vaccination coverage

In bivariate analysis, there were significant associations between full vaccination status and the type of respondent, level of education of mother/caregiver, maternal healthcare decision-making, number of ANC attendances, number of tetanus vaccinations, post-natal care attendance, having information on vaccination and vaccine-preventable diseases, aim for vaccination, number of vaccine-preventable diseases known by the respondent, age

**Table 3 Factors associated with full vaccination coverage of children aged 12–23 months in Demba Gofa District, Southern Ethiopia, 2019, (N = 657).**

| Variables | Category | Full vaccination | | Odds ratio (95% CI) | |
|---|---|---|---|---|---|
| | | Yes | No | COR | AOR |
| Respondent | Mother | 289 (93.5) | 343 (98.6) | 0.21 (0.08, 0.57) | 0.54 (0.13, 2.12) |
| | Caregiver | 20 (6.5) | 5 (1.4) | 1 | 1 |
| Educational level of the respondents | Illiterate | 114 (36.9) | 146 (42.0) | 1 | 1 |
| | Read & write | 29 (9.4) | 14 (4.0) | 2.65 (1.34, 5.25) | 2.00 (0.72, 5.54) |
| | Primary school (1–8) | 75 (24.3) | 98 (28.2) | 0.98 (0.66, 1.44) | 1.03 (0.60, 1.75) |
| | Secondary & prep. (9–12) | 57 (18.4) | 63 (18.1) | 1.16 (0.75, 1.78) | 0.99 (0.57, 1.73) |
| | Diploma & above | 34 (11.0) | 27 (7.8) | 1.61 (0.92, 2.82) | 0.87 (0.41, 1.81) |
| Respondents healthcare decision making | Jointly with husband | 227 (73.5) | 246 (70.7) | 1.71 (1.09, 2.68) | **1.88 (1.06, 3.34)*** |
| | By herself | 47 (15.2) | 37 (10.6) | 2.36 (1.30, 4.27) | **4.03 (1.66, 9.78)*** |
| | Husband alone | 35 (11.3) | 65 (18.7) | 1 | 1 |
| Number of ANC attendance | Once | 9 (3.2) | 19 (7.0) | 1 | 1 |
| | Twice | 40 (14.4) | 78 (28.6) | 1.08 (0.45, 2.61) | 1.26 (0.31, 5.07) |
| | Three times | 111 (39.9) | 82 (30.0) | 2.85 (1.23, 6.64) | 2.64 (0.66, 10.43) |
| | Four times | 118 (42.2) | 94 (34.4) | 2.65 (1.14, 6.12) | 2.22 (0.54, 9.00) |
| Number of TT vaccination | 1 | 30 (11.6) | 49 (21.1) | 1 | 1 |
| | 2 | 133 (51.6) | 124 (53.4) | 1.75 (1.04, 2.93) | 1.31 (0.70, 2.47) |
| | 3 | 71 (27.5) | 42 (18.1) | 2.76 (1.52, 4.99) | 2.00 (0.94, 4.23) |
| | 4 | 24 (9.3) | 17 (7.3) | 2.30 (1.06, 4.98) | 0.99 (0.33, 2.98) |
| PNC attendance | Yes | 46 (14.9) | 17 (4.9) | 3.40 (1.90, 6.08) | **5.02 (2.28, 11.05)*** |
| | No | 263 (85.1) | 331 (95.1) | 1 | 1 |
| Information on vaccination & VPDs | Yes | 302 (97.7) | 307 (88.2) | 0.17 (0.07, 0.39) | 0.71 (0.16, 3.11) |
| | No | 7 (2.3) | 41 (11.8) | 1 | 1 |
| Objective of vaccination | To prevent disease | 235 (76.1) | 209 (60.1) | 4.09 (2.04, 8.17) | 3.44 (0.85, 13.81) |
| | For child health | 63 (20.4) | 99 (28.4) | 2.31 (1.10, 4.84) | 1.88 (0.47, 7.52) |
| | Do not know | 11 (3.6) | 40 (11.5) | 1 | 1 |
| Number of VPDs known by the respondent | Single disease | 90 (29.1) | 115 (33.0) | 1.48 (0.99, 2.19) | 0.90 (0.51, 1.60) |
| | More than one disease | 145 (46.9) | 93 (26.7) | 2.95 (2.01, 4.33) | **1.82 (0.99, 3.34)*** |
| | Do not know | 74 (23.9) | 140 (40.2) | 1 | 1 |
| Age at which a child begins vaccination | Just after birth | 155 (50.2) | 132 (37.9) | 4.69 (2.26, 9.75) | 0.82 (0.19, 3.60) |
| | 1 month after birth | 144 (46.6) | 176 (50.6) | 3.27 (1.58, 6.77) | 0.65 (0.14, 2.87) |
| | Do not know | 10 (3.2) | 40 (11.5) | 1 | 1 |
| Sessions needed to complete vaccination | Three & less | 48 (15.5) | 64 (18.4) | 1.72 (0.97, 3.04) | 0.42 (0.16, 1.11) |
| | Four or five | 212 (68.6) | 183 (52.6) | 2.66 (1.66, 4.27) | 0.74 (0.33, 1.67) |
| | Six & above | 19 (6.1) | 32 (9.2) | 1.36 (0.67, 2.78) | 0.34 (0.12, 1.00) |
| | Do not know | 30 (9.7) | 69 (19.8) | 1 | 1 |
| Age to complete vaccination | Before 1 year | 192 (62.1) | 202 (58.0) | 2.51 (1.55, 4.05) | 0.92 (0.40, 2.09) |
| | 1 year & above | 89 (28.8) | 72 (20.7) | 3.26 (1.91, 5.57) | **2.54 (1.04, 6.21)*** |
| | Do not know | 28 (9.1) | 74 (21.3) | 1 | 1 |
| Vaccination will not make child to sick | Yes | 275 (89.0) | 280 (80.5) | 1.96 (1.26, 3.06) | **0.32 (0.16, 0.64)*** |
| | No | 34 (11.0) | 68 (19.5) | 1 | 1 |

**Note:**
**Key** * *p*-value < 0.05, COR, Crude odds ratio, AOR, Adjusted odds ratio.

at which a child began vaccination, sessions needed to complete vaccination, age at which a child completed vaccination, and whether a vaccination made a child sick or not.

In multivariable analysis, maternal healthcare decisions made jointly with husbands (AOR = 1.88, 95% CI [1.06–3.34]), maternal healthcare decision made by mothers

(AOR = 4.03, 95% CI [1.66–9.78]), number of post-natal care attendances (AOR = 5.02, 95% CI [2.28–11.05]), age at which the child completed vaccination (AOR = 2.54, 95% CI [1.04–6.23]), and vaccination not making the child sick (AOR = 0.32, 95% CI [0.16–0.64]) were all significantly associated with full vaccination status (Table 3).

## DISCUSSION

This study measured full vaccination coverage and associated factors among children aged 12–23 months in Demba Gofa District, Southern Ethiopia. In this study, only 47.0% of children were fully vaccinated. BCG, OPV3, pentavalent 3, PCV3, and Rota had a similar coverage of 88.7%, but measles coverage was only 59.4%, far below the others. Compared to the mini EDHS 2019 report, our results show slightly higher coverage (*Central Statistical Agency: Addis Ababa Ethiopia, 2019*) but equivalent coverage to results from Mizan Aman town (*Meleko, Geremew & Birhanu, 2017*) and Arba Minch Zuriya District (*Facha, 2015*) in Southern Ethiopia.

However, our coverage was lower than those reported for Lay Armachiho District, North Gondar Zone (*Kassahun, Biks & Teferra, 2015*) and Sinana District, Southeast Ethiopia (*Legesse & Dechasa, 2015*). Furthermore, coverage was very low compared to coverage reported for Tehulederie District, northeast Ethiopia (97%) (*Ebrahim & Salgedo, 2015*), Debre Markos town, Amhara regional state (92%) (*Gualu & Dilie, 2017*), and studies conducted in Cameroon (*Russo et al., 2015*) and South Nigeria (*Adeleye & Mokogwu, 2016*). The differences discripancy may be due to differences in socioeconomic factors like residence and relatively poor healthcare systems as well as the settings in which the studies were conducted.

Different factors were examined with respect to associations with the full vaccination status. No socio-demographic characteristic was significantly associated with the full vaccination status of a child. This is in contrast to other studies conducted in different parts of Ethiopia (Arbegona district, South Ethiopia; Mizan Aman town, Bench Maji Zone, Southwest Ethiopia; Mecha District, Northwest Ethiopia; Jigjiga district, Somali national regional state, and Addis Ababa city (*Negussie et al., 2016*; *Meleko, Geremew & Birhanu, 2017*; *Debie & Taye, 2014*; *Mohamud et al., 2014*; *Birhanu et al., 2016*)) which have shown that maternal age, maternal level of education, maternal occupation, and husbands' level of education were significantly associated with full vaccination coverage. A study from Bangladesh also showed that maternal age and maternal level of education were significantly associated with full vaccination coverage. The possible reasons for these differences may be a due to differences in the study designarea, other sociodemographic characteristics of the participants, and the study settings.

We also found no associations between the children's characteristics and full vaccination coverage. However, studies from Arbegona District, Southern Ethiopia; Debre Markos town, Amhara regional state; Indonesia; Bangladesh; Sinana District, Southeast Ethiopia; and Addis Ababa town (*Negussie et al., 2016*; *Gualu & Dilie, 2017*; *Holipah, Maharani & Kuroda, 2018*; *Sheikh et al., 2018*; *Legesse & Dechasa, 2015*; *Birhanu et al., 2016*) all showed that age, sex, and birth order of the children were significantly associated with full vaccination coverage.
Mothers who made healthcare decisions jointly with their husbands were 1.88 (95% CI [1.06–3.34]) times more likely to vaccinate their children fully than when decisions were made by the husbands alone. Furthermore, mothers who made healthcare decisions themselves were 4.03 (AOR 95% CI [1.66–9.78]) times more likely to fully vaccinate their children than when decisions were made by husbands alone. This is similar to a study from Indonesia (*Herliana & Douiri, 2018*) and indicates that maternal healthcare decisions made jointly with husbands and by mothers alone give mothers more freedom of health choices that benefit their children.

Mothers who attended post-natal care follow-up were 5.02 (AOR 95% CI [2.28–11.05]) times more likely to fully vaccinate their children than mothers who did not. This is similar to a study from East-central Ethiopia (*Yenit, Gelaw & Shiferaw, 2018*). This may be because during postnatal care follow-up, mothers have a greater chance to obtain advice on vaccinations and exposure to the service that increases the probability of being fully vaccinated.

Respondents who replied that a child should complete vaccination by 1 year of age were 2.54 (95% CI [0.04–6.23]) times more likely to fully vaccinate their children than those who did not know the exact age of vaccination completion. This is similar to studies from the Lay Armachiho District, North Gondar Zone, Northwest Ethiopia; Ambo District, Central Ethiopia; and Addis Ababa city (*Aregawi et al., 2017*; *Etana & Deressa, 2012*; *Birhanu et al., 2016*). A possible explanation may that having better knowledge on the timing of vaccination affected utilization of the service, as improved knowledge is expected to positively influence the uptake and utilization of a healthcare service.

Mothers who responded that vaccination will not make the child sick were less likely to vaccinate their children fully than those who thought that it would. Reasons given by respondents who did not complete vaccination were vaccinator absenteeism, a lack of awareness, time inconvenience, and not knowing the vaccination date. This was similar to a study from Laelay Adiabo District, Tigray region, Northern Ethiopia (*Aregawi et al., 2017*). Furthermore, feeling that the vaccination had no use, a fear of side effects, religious and cultural refusals, a belief that vaccination hurts the child, and a lack of awareness were reasons raised by mothers/caregivers who did not vaccinate their children at all. Similar reasons have been given in other studies (*Gualu & Dilie, 2017*; *Yenit, Gelaw & Shiferaw, 2018*; *Aregawi et al., 2017*).

The strength of this study are applying community-based study in a remote and rural part of the country and using a simple random sampling technique. However, recall bias of mothers could affect the study findings dispite the efforts made to minimize them by asking the injection site, repepitions of the shcedules, looking at the scars. Vaccine supply management, which could affect vaccination uptake, was not assessed.

## CONCLUSIONS

Less than half of children studied here were fully vaccinated and the prevalence of unvaccinated children was 11.3%, far below the government target of 90% necessary for sustained control of vaccine-preventable diseases. Maternal healthcare decision-making, postnatal care attendance, the objective of vaccination, the age at which a child begins and

completes vaccination, and sessions needed to complete vaccination were significant factors associated with full vaccination. Absenteeism of vaccinators, a lack of awareness, time inconvenience, not knowing the exact date of vaccination, and local religious and cultural contexts were reasons given for not fully vaccinating their children.

We advise that district health offices continuously conduct systematic supervision and periodic evaluation of vaccination performance of health facilities using a standardized checklist. Also, there should be an adequate supply of vaccination cards to health facilities (health centers and health posts). The health sector and other legal bodies are advised to improve maternal healthcare decision-making by empowering women on decision-making processes. Promoting institutional delivery and postnatal care services could improve full vaccination coverage. Health extension workers should emphasize explaining the specific names of vaccine-preventable diseases, the age at which a child begins and completes vaccination, and the number of sessions needed to complete vaccinations. Starting with senior management and extending to health extension workers, there is a need for improved planning of convenient times for the parent to attend vaccination, evaluating performance among target children, estimating the frequency of missed children, and providing the opportunity for re-attendance for defaulting children. The vaccine supply chain should also be studied further.

## ACKNOWLEDGEMENTS

Our thanks extend to district health managers and staff for their support in facilitating the research process by timely approving and writing support letters. We would also like to thank Serawit Samuel for reviewing statistical part of this study. Finally, special thanks go to data collectors and study participants who contributed to this study.

### Funding

The authors received no funding for this work.

### Competing Interests

The authors declare that they have no competing interests.

### Author Contributions

- Tadele Dana Darebo conceived and designed the experiments, performed the experiments, analyzed the data, prepared figures and/or tables, authored or reviewed drafts of the paper, and approved the final draft.
- Bahru Belachew Oshe conceived and designed the experiments, performed the experiments, analyzed the data, prepared figures and/or tables, authored or reviewed drafts of the paper, budget contribution, and approved the final draft.
- Chala Wegi Diro performed the experiments, analyzed the data, prepared figures and/or tables, authored or reviewed drafts of the paper, and approved the final draft.

## Human Ethics

The following information was supplied relating to ethical approvals (*i.e.*, approving body and any reference numbers):

Wolaita Sodo University college of health sciences and medicine ethical review committee approved this study.

## Data Availability

The raw data is available in the Supplemental File.

## Supplemental Information

Supplemental information for this article can be found online at http://dx.doi.org/10.7717/peerj.13081#supplemental-information.

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
