# Peer review of "Full vaccination coverage and associated factors among children aged 12 to 23 months in remote rural area of Demba Gofa District, Southern Ethiopia"

_PeerJ, doi:10.7717/peerj.13081_

## Round 0.1 · original submission · Major Revisions

We invite you to consider carefully the comments given below and to submit a fully revised manuscript incorporating changes, and including any new experiments that you consider necessary to address all the issues raised by the reviewers.

In particular, please note the major methodological concern, which is that all of the results are based on recall in 657 children which affects the validity of the findings.

Please explain how mothers remembered vaccination status if there are no records? The authors need to show how and why all the children they studied did not have vaccination cards and what efforts were made to reduce recall bias.

·

Basic reporting

This is a well written, potentially useful manuscript with a fascinating theme. The introduction is concise with previous findings and rationale included. The methods are appropriate but require a few clarifications. The results and discussion are clear and compelling. Overall, this is a good quality manuscript that has implications for childhood vaccination in Ethiopia. I think some parts of the manuscript can be improved. Below are many suggestions.

Introduction
[1] Lines 52-58 talk about recommended vaccines by WHO, but the authors did not elaborate on the vaccination programme in Ethiopia. The authors did not make clear what the EPI is and what it entails.
[2] EPI ends at nine months so, why was your study group aged up to 23 months? Moreover, are you considering only vaccines on EPI? Then it would help if you made it known.
[3] Authors need to give more epidemiological statistics on vaccination issues in Ethiopia; for example, the rate of fully, partially, and unvaccinated children in Ethiopia; the prevalence of vaccine-preventable disease; how many still die from vaccine-preventable disease or suffer from chronic morbidity related to these diseases in Ethiopia.
[4] Lines 69-70 mentioned the vaccination rate of 39 to 43%. Does it stand for fully or partially vaccinated children?

Experimental design

Methods
[5] In line 99, why did you use the prevalence of full immunisation from a single study by Etana and Deressa in 2012 (N= 536) instead of the pooled estimate from recent systematic reviews and meta-analyses? Also, I think it is good to include these recent systematic reviews in your paper.
- https://doi.org/10.1186/s12889-020-09118-1
- https://doi.org/10.1016/j.vaccine.2017.04.034

[6] Line 108: what do the authors mean by total sample size was divided by the size of all households? Is it the number of house occupants? If yes, does it mean that your denominator contained participants not eligible for the study?
[7] Lines 109-110: what do authors mean here?
"Then, children were proportionally allocated according to the number of children they had".
[8] Your sampling technique needs to be revised. Is it not better to use a cluster sampling technique?
[9] Did a qualified translator translate the questionnaire? If no, did the translation have the same meaning?
[10] Lines 117 – 118, you mentioned that the questionnaire was developed from the literature. How was this developed questionnaire assessed for validity and reliability?
[11] In lines 126 – 127, why was pretesting performed on 5% of the sample size? Please mention a reference for this.
[12] In lines 30 – 34, it is important to mention a reference for the definitions of fully vaccinated, partially vaccinated, and unvaccinated.
[13] In lines 135-138, were you quantifying the knowledge and attitudes? If yes, is this a valid knowledge score and attitude score? A reference should be included.
[14] Lines 120-121 talk about training, but the training was given to who? By who? And what for? Is it the same training as in line 125?
[15] Why were interviewer-administered questionnaires used instead of self-completed questionnaires (mothers asked to show vaccination cards)? Would this introduce social-desirability bias?
[16] In lines 146 – 152, what was the dependent variable in the bivariate and multivariate analysis? In a previous study by Etana and Deressa, "full immunisation status of the children (card plus mothers recall) was included in the logistic regression model as a dependent variable".
[17] Was a chi-square test not necessary in your analysis?

Validity of the findings

Results
[18] I think the authors should include the relevant responses for the questions in lines 185-187.
[19] The knowledge and attitude score in lines 187-189. More details need to be provided. See comment [13].
[20] The statement in lines 192-193 is contradictory to lines 87-89 which states that EPI service (where I suppose vaccination is done) was provided only by governmental health facilities and not health posts. Are health posts also government health facilities. Why is the EPI service provided only by governmental health facilities?
[21] Lines 197 – 198, you stated that "No child had a vaccination card, and all data were collected by recall alone". Why? Is it standard practice in this area? Are vaccination cards kept in the government health facilities?
Etane and Deressa showed a discrepancy between full vaccination by card only (27.8%) and full vaccination by card and recall (35.6%). How reliable is your finding "47.0% were fully vaccinated or completed all the recommended vaccines before celebrating their first birthday"? Lines 274 – 275, you stated that "recall bias of mothers could affect the study findings dispite the efforts made to minimise them". What were the efforts made?
[22] Lines 229-230 and lines 238-239: only general reasons are given for the differences between your results and the one in other studies. Authors need to provide precise and concise explanations for that.
[23] In lines 240-244, why was there a difference between your study and others?

Additional comments

Conclusion
[24] The conclusion needs to be summarised and must be based on the results

General
[25] The English language and the flow would need to be verified.

·

Basic reporting

Vaccination is the administration of a biological substance in (line 43) might not be appropriate since we have lots of biological substances with classifications. I would advise you to use simple terms “administering vaccines” or you could say “vaccination is the efficient way to protect people from infectious diseases”.
In line 45 you could go straight to the point by stating “vaccination has reduced and eliminated the six childhood killer diseases”. I do not think it is necessary to mention all the diseases.
Again, in line (48-50), you could be more concise. “Global vaccination coverage has increased, contributing to decreases in child mortality from 9.6 million in 2000 to 5.9 million in 2015, especially low and middle-income countries”.
Have a look at line (57-60) “Expanded Program on Immunization (EPI) agenda plans to give the primary vaccination series to at least 90% of children”. I suggest you put a full stop here. Start another sentence with “Despite these efforts, the target has still to be attained in many LMICs”. You can now continue to write “For instance”, “Over 20,000 children…….”. I will suggest you write the word twenty Thousand in line 59 in figures 20,000. It conveys a message, and it stands out.
Also, you could remove “only in only ten countries” in line 66. Just write “in only 10 countries including Ethiopia”.
Line 68-70 need referencing
Can you be more formal in lines 70-71? I suggest you write “reported to be low (15%) in Afar and High (89%) in Addis Ababa”. The use of “reported to be as low as 15% in Afar and as high as 89% in Addis Ababa” seems inappropriate.

Experimental design

Since the questionnaire was developed by the authors, it means that it has not been applied elsewhere, How did you check for validity and reliability of the questionnaire during pre-testing
Under the Study design, I suggest you explain why you chose Demba Goaf District for the study. Are there frequent outbreaks in this district? Have reviewed records shown that this district is trailing under the EPI program? Tell us the gap you are going to fill in this community.
Also, give reasons why you chose to work on the age group 12-23 months.
Why do you choose to exclude mothers who are mentally disabled/critically ill and caretakers? What are the reasons for excluding them?
What is the difference between caregivers and caretakers? Because caretakers ware in your exclusion criteria and caregivers have been mentioned in lines 121-123 during data collection

Validity of the findings

In lines, 88-89 Is it only government institutions that provide EPI in Ethiopia? No other facilities provide EPI programs.
How did you come by Pre-testing of 5%? Is it standard?
In lines 197-198 how come all 657 children had no vaccination cards? Is it possible that all these populations of children did not have vaccination records? I doubt the validity of the results with only recall.
How did you check for recall bias since you stated this in lines 274-275. Could you explain the measures put in place to check recall bias?

Additional comments

Overall the study is important

---

## Round 0.2 · Minor Revisions

As requested from Reviewer 1, please ask a biostatistician to review your statistical analysis, then you can acknowledge him/her in your manuscript. Also, please take some minor comments from Reviewer 2 into account.

·

Basic reporting

No additional comment

Experimental design

No additional comment

Validity of the findings

No additional comment

Additional comments

Data analysis and results should be reviewed by a biostatistician before publication.
You have adequately addressed the concerns and the information will add to the general literature.

·

Basic reporting

I commend the authors for making the necessary changes to clear all ambiguity.

Experimental design

Thank you for addressing the necessary concerns. However, since the vaccination cards serve as a form of reminder to many mothers and caregivers, and one of the reasons for this study, it will be appropriate to suggest it as your first policy recommendation.

Validity of the findings

No comments

Additional comments

Overall, the concerns raised have been addressed

---

## Round 0.3 · accepted · Accept

Please take some minor comments about language from Reviewer 1 into account.

·

Basic reporting

I have no further comments.

Experimental design

I have no further comments.

Validity of the findings

I have no further comments.

Additional comments

The authors have responded to reviewers comments and the study is clear and
addresses an important research question.
The English language should be improved to ensure that an international audience can clearly understand your text. An audience might find some of these sentences hard to read. For example
Lines 58-63
Lines 92-95
Lines 136-142
Lines 222-228

·

Basic reporting

No comments

Experimental design

No comments

Validity of the findings

No comments

Additional comments

No comments